# Powdery Mildew Resistance Genes in Vines: An Opportunity to Achieve a More Sustainable Viticulture

**DOI:** 10.3390/pathogens11060703

**Published:** 2022-06-18

**Authors:** Viviana Sosa-Zuniga, Álvaro Vidal Valenzuela, Paola Barba, Carmen Espinoza Cancino, Jesus L. Romero-Romero, Patricio Arce-Johnson

**Affiliations:** 1Departamento de Genética Molecular y Microbiología, Facultad de Ciencias Biológicas, Pontificia Universidad Católica de Chile, Avenida Libertador Bernardo O’Higgins 340, Santiago 8331150, Chile; vasosa@uc.cl; 2Facultad de Agronomía e Ingeniería Forestal, Pontificia Universidad Católica de Chile, Avenida Vicuña Mackenna 4560, Santiago 7820436, Chile; 3Foundazione Edmund Mach, Via Edmund Mach 1, San Michele all’Adige (TN), 38010 Trento, Italy; alvaroignaciovidalvalenzuela@gmail.com; 4Instituto de Investigaciones Agropecuarias, Avenida Santa Rosa 11610, Santiago 8831314, Chile; paola.barba@inia.cl; 5Instituto de Ciencias Biomédicas, Facultad de Ciencias de la Salud, Universidad Autónoma de Chile, Avenida El Llano Subercaseaux 2801, Santiago 8900000, Chile; carmen.espinoza.cancino@gmail.com; 6Departamento de Biotecnología Agrícola, Instituto Politécnico Nacional, Centro Interdisciplinario de Investigación para el Desarrollo Integral Regional, Unidad Sinaloa, Bvd. Juan de Dios Bátiz Paredes 250, Culiacan Rosales 81101, Mexico; jromeror@ipn.mx; 7Agrijohnson Ltda., Parcela 16b, Miraflores, Curacavi 9630000, Chile

**Keywords:** *Erysiphe necator*, grapevine, resistance genes, *Run*, *Ren*, powdery mildew

## Abstract

Grapevine (*Vitis vinifera*) is one of the main fruit crops worldwide. In 2020, the total surface area planted with vines was estimated at 7.3 million hectares. Diverse pathogens affect grapevine yield, fruit, and wine quality of which powdery mildew is the most important disease prior to harvest. Its causal agent is the biotrophic fungus *Erysiphe necator*, which generates a decrease in cluster weight, delays fruit ripening, and reduces photosynthetic and transpiration rates. In addition, powdery mildew induces metabolic reprogramming in its host, affecting primary metabolism. Most commercial grapevine cultivars are highly susceptible to powdery mildew; consequently, large quantities of fungicide are applied during the productive season. However, pesticides are associated with health problems, negative environmental impacts, and high costs for farmers. In paralleled, consumers are demanding more sustainable practices during food production. Therefore, new grapevine cultivars with genetic resistance to powdery mildew are needed for sustainable viticulture, while maintaining yield, fruit, and wine quality. Two main gene families confer resistance to powdery mildew in the Vitaceae, *Run* (Resistance to *Uncinula necator*) and *Ren* (Resistance to *Erysiphe necator*). This article reviews the powdery mildew resistance genes and loci and their use in grapevine breeding programs.

## 1. Introduction

Grapevine (*Vitis vinifera*) is one of the main fruit crops worldwide. In 2020, the total surface area dedicated to this crop was estimated to be 7.3 million hectares [1,2], with a production of approximately 77.8 million tons of grape clusters. Of the total harvest, 57% is destined for wine production; 36% corresponds to table grapes, and 7% is used to produce raisins [1]. Yield and fruit quality are affected by the attack of different fungal pathogens [3]. Of these, powdery mildew is the most important and challenging preharvest disease due to its high destructive force, the high susceptibility of most commercial cultivars [3,4], and the broad humidity and temperature ranges in which the pathogen thrives and develops [5]. Its causal agent is the biotrophic fungus *Erysiphe necator* (synonyms: *Uncinula necator* Burr) [6,7]. The main symptoms typically associated with infection are decreased cluster weight, delayed fruit ripening, and reduced photosynthetic and transpiration rates, although Pimentel et al. (2021) [8] observed no differences in berry weight, sugars, organic acids, or main ripening parameters between infected and healthy berries. The determination of yield loss caused by powdery mildew attack is difficult to standardize because multiple factors, such as cultivar susceptibility, production system, and moment of infection, are involved [9,10].

Powdery mildew not only affects crop productivity but also has an impact on fruit quality, altering sugar content, acidity level [11], and anthocyanin levels [12]. Moreover, additional negative sensorial effects on wine quality have been described, such as the reduction in vanilla-like aromas in red wines [13] and tropical fruit-like aromas in Sauvignon blanc [12]. Color is yet another parameter influenced by *E. necator* as reductions in the anthocyanin content in fruits diminish the intensity of color in red wines [14].

In addition, powdery mildew induces metabolic reprogramming in its host [8]. At the primary metabolic level, it reduces the abundance of glycolytic, photorespiratory, and photosynthetic proteins [15] and generates a redistribution of carbon reserves due to an increase in invertase and alpha-amylase activity [16], which degrades starch reserves to glucose and maltose [17]. This metabolic alteration is accompanied by an upregulation of the transcription of the hydroxymethyl-flutary-CoA (*HMG-CoA*) and *HMG-CoA* reductase genes [16]. HMG-CoA synthase enzyme converts Acetoacelyl-CoA into 3-hydroxy-3-methylglutaryl-CoA, which is transformed into mevalonate by HMG-CoA reductase. Both molecules are part of the biosynthesis pathway of terpenes, carotenoids, and sterol compounds [18].

Most commercial grapevine cultivars are highly susceptible to *E. necator* [19]. For that reason, in order to achieve stable yields and good-quality fruits, powdery mildew is controlled by the intensive application of fungicides during the productive season [20]. However, chemical control is expensive for farmers and is associated with health hazards for field workers, animals, and consumers of table grapes and wine [20,21,22,23,24]. In addition, fungicide application has negative consequences on the environment, such as soil and groundwater contamination [25]. In response to these detrimental effects, governments and consumers are demanding more sustainable production methods, including decreased pesticide applications [26]. One example of this is the Green Deal Farm Fork strategy, which aims to reduce the use of pesticides in Europe by 50% [27]. These demands and legislations are a great challenge for viticulture farmers, who, at the same time, are facing the effects of climate change that threaten the yield and quality of their production [26]. In this context, the approach of replacing conventional grapevine cultivars with fungus-resistant cultivars is a sustainable alternative for disease control [28].

Fungal-resistant cultivars can be developed both by traditional genetic improvement, using directed crosses with species from related botanical families that naturally carry resistant loci and by genome editing [28]. This review summarizes the use of *Run* (Resistance *to U. necator*) and *Ren* (Resistance *to E. necator*) gene families that confer resistance to powdery mildew in grapevines (*V. vinifera*) and the host’s response to infection. In addition, a summary of the current advances in the development of resistance to powdery mildew by gene editing is presented and discussed.

## 2. Host Response

The plant’s immune system is summarized by the zig-zag model, which distributes the plant’s response to the presence of pathogens into three main stages. The initial stage is related to the recognition of Pathogen-Associated Molecular Patterns (PAMPs) or Microbe-Associated Molecular Patterns (MAMPs) by Pattern Recognition Receptors (PRRs), resulting in PAMP-triggered immunity (PTI). This triggers nonspecific physiological and molecular responses, such as the accumulation of reactive oxygen species (ROSs) and phytoalexins, and/or stomata closure through the phosphorylation of a MAP kinase pathway (MAPKKK–MAPKK–MAPK), which activates transcription factors, such as WRKY22, thus inducing related genetic responses. In the second stage, and in response to plant defense, pathogens initiate effector-triggered susceptibility (ETS) [29] whereby through effectors (Secreted Effector Proteins), such as coat proteins [30] or other specific proteins, the defensive response pathway of plants is stopped. For instance, some effectors, such as AvrPto and AvrPtoB, have been shown to block the phosphorylation of MAPKs in the case of *Pseudomonas syringae* [31], while *EqCSEP01276* produced by powdery mildew inhibits the biosynthesis of abscisic acid (ABA) [32]. In this ETS phase, pathogens may overcome the immune response of plants and infect the host’s cells. Cells of certain plant species possess resistance proteins (R) that directly or indirectly recognize the presence of pathogenic effectors and trigger an immune response, called effector-triggered immunity (ETI). This final phase generates an immune response of greater intensity than PTI. This switch between ETS–ETI is maintained until the hypersensitive cell death response is triggered or the pathogen overwhelms the cell [33].

Most *R* genes encode nucleotide-binding site (NBS) leucine-rich repeat (LRR) domain proteins (NBS–LRR proteins) [29]. This is the case of the *R* genes transcribed in the Vitaceae plant family in response to *E. necator* infection. In Vitaceae, the *R* genes are clustered in tandem repeats of genomic regions. These have been genetically mapped, uncovering nine loci that encode *R* gene sequences conferring resistance to *E. necator*, such as *Run1*, *Run2*, *Ren1*, *Ren2*, *Ren3*, *Ren4*, *Ren5*, *Ren6,* and *Ren7* [34], which have been used to obtain plants resistant to this infection by pseudo-backcrossing [35]. On the other hand, more recent “New Breeding Technologies” (NBTs) have been employed for genetic improvements in *Vitis* plants through the elimination of the endogenous genetic material using the thermal shock FRP/FLP system [36,37] or the generation of DNA-free modifications using ribonucleoproteins [38]. This, together with new rapidly developing *Vitis* models, such as Microvine or Picovine, have helped to accelerate the discovery of new target genes to decipher the resistance of *Vitis* to powdery mildew, such as the PATHOGENESIS-RELATED 4b (*VvPR4b*) gene, whose loss of function decreases *Vitis* resistance to downy mildew [39]. As expected, the overexpression of *VvPR4b* is related to enhanced resistance to *E. necator* [40], while the DIMERIZATION PARTNER-E2F-LIKE 1 (*VviDEL1)* double-cut transgenic *Vitis* has 90% fewer symptoms of powdery mildew infection than the control plants [40].

Hormones play a key role in plant defense responses, particularly jasmonic acid (JA) and ethylene (Et) for necrotrophic pathogens and salicylic acid (SA) for hemibiotrophic and biotrophic pathogens, such as powdery mildew [3,8]. In *Arabidopsis thaliana*, SA is synthesized in response to a pathogen attack, mainly from chorismic acid by the activity of the enzymes isochorismate synthase (ICS) and isochorismate pyruvate lyase (IPL) [41]. A mobile derivative of SA is methyl salicylate (MeSa), which can be transported through the phloem to distal parts of plants, generating a Systemic Acquired Response (SAR). This activates various physiological immune responses, such as programmed cell death (PCD) and accumulation of ROS, such as hydrogen peroxide and nitric oxide [42]. Thus, to achieve an effective resistance response in grapevines upon infection by *E. necator*, it is necessary to enhance SAR [3]. Although the most well-described hormonal response pathway against the attack of powdery mildew is that of SA, it has also been shown that Et and JA contribute to the response against *E. necator* in grapevines [43,44]. Furthermore, recent data show that when *V. vinifera* cv. ‘Cabernet Sauvignon’ plants are treated with exogenous Et, a defense response against *E. necator* is triggered [44]. Such a response mechanism is associated with the induction of a series of defense proteins, such as acidic class IV chitinase (CHIT4c), protease inhibitor (PIN), polygalacturonase-inhibiting protein (PGIP), and ß-1,3-glucanase (GLU). Although there is no direct evidence linking the induction of these defense proteins with the phenylpropanoid pathway, a correlation has been seen in the increased biosynthesis of phytoalexins and the upregulation of phenylalanine ammonia-lyase (*PAL*) and stilbene synthase (*STS*) genes. These increases are positively correlated with the increased accumulation of stilbenes with known antimicrobial activity, which emphasizes the participation of these enzymes in the host response against biotrophic fungi [45]. In support of the above, the transcriptomic analysis of the response to *E. necator* infection of two *Vitis* species, one susceptible (*V. pseudoreticulata*) and the other resistant (*V. quinquangularis*), showed the induction of genes and metabolites associated with the defense response [46]. Specifically, the repression of the flavonoid pathway genes was reported in the susceptible cultivar *V. pseudoreticulata*, alongside differential responses of genes and processes related to hormones, such as SA and JA [47]. A high accumulation of arachidic acid has been reported in berries infected by *E. necator*, meaning that it is now considered a quantitative biomarker for infection by this fungus [8,48]. Interestingly, Jiao et al. [46] described the suppression of genes related to the biosynthesis and elongation of fatty acids in the resistant cultivar, suggesting the participation of these types of lipids in the interaction of *E. necator* with the host in a developing infection. Additionally, genes involved in the biosynthesis and signaling of phytohormones, such as JA and cytokinins (CK), were identified, as were ones that code for protein kinases and proteins with NBS–LRR repeats [46].

## 3. Mapping Resistance Genes for Powdery Mildew Resistance Using Interspecific Crosses

The use of F_1_ families derived from the cross of two parents with contrasting phenotypes is the most used strategy for genetic mapping in grapevines [49]. Based on the pseudo-testcross strategy [50], it is suitable for highly heterozygous plants with long juvenile periods, such as grapevines.

Although *V. vinifera* is the most widely cultivated *Vitis* species, the levels of powdery mildew resistance in this species are lower than that of other *Vitis* or *Muscadinia* species from North America or Asia. These contrasting phenotypes have been exploited for genetic mapping. To date, 15 loci responsible for grapevine powdery mildew resistance have been reported, leveraging information from 24 F_1_ interspecific families or descendants [51].

Strong disease-resistant loci have been mapped to chromosomes 12, 18, and 9, named *Run1* [52,53], *Ren4* [54,55,56], and *Ren6* [57], respectively. These loci originate from *M. rotundifolia*, *V. romanetii*, and *V. piasezkii* and provide strong quantitative disease resistance [58].

Other moderate to minor disease-resistant sources have been found on chromosomes 2, 9, 13, 14, 15, and 18 [51]. In some of these loci, the study of the infection process demonstrated a postpenetration resistance mechanism, with delayed hyphal growth, as in the case of *Ren1* [59], *Ren5* [60], *Ren7* [57], and *Ren11* [61,62]. Some of these moderate to minor resistance loci come from *V. vinifera*, *V. rotundifolia* [60], *V. piasezkii* [57], and complex hybrids involving *V. cinerea*, *V. rupestris*, or ‘Seibel’ selections [63,64,65,66,67].

## 4. *Run* and *Ren* Resistance Genes

Several loci associated with powdery mildew resistance have been identified in different species of the Vitaceae family. These loci have been named *Ren1* [59], *Ren1.2* [68], *Ren2* [63,69], *Ren3* [65,70], *Ren4* [54], *Ren5* [60], *Ren6* [57], *Ren7* [57], *Ren8* [66], *Ren9* [65], R*en10* [67], *Ren11* [61], *Run1* [52,53], *Run1.2a* and *b* [71], *Run2.1* [55,71], and *Run2.2* [55,71] (Figure 1). In the case of most *Run* and *Ren* loci, it is not clear which genes are responsible for powdery mildew resistance and their mechanism of action [34]. The only exception to this is the resistance gene *MrRUN1* (*MrRGA10*), whose sequence was described by Feechan et al. (2013) [53]. The *MrRUN1* gene encodes an NBS–LRR resistance protein containing a Toll/interleukin-1 receptor-like (TIR) domain, which recognizes pathogen effectors, thus triggering the hypersensitive response (HR), which is characterized by an increase in ROS production leading eventually to programmed cell death (PCD) in infected cells [69]. The same defense response has been seen in grapevine plants that carry the *Run1*, *Run1.2a*, *Run1.2b*, *Run2*, *Ren1*, *Ren2*, *Ren3*, *Ren4*, *Ren5*, *Ren6*, *Run7*, or *Ren9* loci (Table 1). These facts suggest that the immune response generated by these loci is mediated by resistance proteins that recognize *E. necator* effectors and activate ETI [34]. This hypothesis is supported by the presence in other species of resistance genes to powdery mildew that encode for NBS–-LRR proteins [72,73,74,75,76,77,78,79].

For example, in wheat (*Triticum* spp.), several powdery mildew (*Pm*) genes that encode NBS–LRR proteins have been described. These genes confer a broad-spectrum or a race-specific or a quantitative resistance to the host. Further, their expression could change depending on the plant’s phenological stage. For example, *Pm21* gene encodes an NBS–LRR protein that confers broad-spectrum resistance to powdery mildew (*Blumeria graminis* f.sp. *tritici*) throughout the life of the plant [75]. On the other hand, *Pm6* and *Pm8* genes confer a race-specific resistance that is only present during the adult stage of plant development [77]. One example of quantitative resistance is the Reaction to Puccinia recondite Rob. ex Desm. 22a (*LRR22a*) gene that gives a quantitative resistance at the adult stage of the plant [79].

Another example is the presence of NBS–LRR resistance to powdery mildew (*Sphaerotheca pannosa*) genes in chestnut rose (*Rosa roxburghii* Tratt.). Xu et al. [78] identified and cloned 23 NonTIR–NBS–LRRR and 11 TIR–NBS–LRR genes associated with powdery mildew resistance.

It is important to consider that NBS–LRR resistance proteins confer a level of response that can vary depending on the allele, environmental conditions, and pathogen genotype, an example of which is the race-specific performance of some *Run* and *Ren* loci (Table 1).

In the last decade, the information related to the *Run* and *Ren* resistance loci has increased rapidly. For this reason, a summary of the existing information regarding each locus is presented in the following subsections.

### 4.1. Run1, Run1.2a, and Run1.2b

*Run1* was one of the first *E. necator r*esistance loci described [80]. Despite this, more studies are still needed to elucidate its mechanism of action in more detail.

Among the immune responses triggered by *Run1* in resistant plants is the rapid programmed cell death (PCD), which prevents the development of secondary hyphae and sporulation [35,69]. In the case of cells where the fungus formed secondary hyphae, a fast HR at 48 h postinfection (hpi) is observed by the increase in ROS and the occurrence of PCD [4,35,53]. Another response produced by *Run*1 is the accumulation of callose deposits at the *E. necator* infection site [35].

*Run*1 was described and named for the first time as the Resistance to *Uncinula necator* 1 (*Run*1) locus by Bouquet et al. [80], who generated a segregating population through pseudo-backcrossing of different cultivars of *V. vinifera* and *Muscadinia rotundifolia* G52 that proved resistant to powdery mildew. They observed that *Run1* resistance segregated as a dominant monogenic trait in the population studied. However, no techniques were available at that time that would have enabled the detection of the presence of the *Run1* locus in the progeny without performing phenotypic evaluations. Nevertheless, Pauquet et al. [81] developed AFLP markers linked to *Run1* resistance, facilitating such studies. Because *Run1* had been defined as a chromosomal region and not as a particular gene, further investigations sought to obtain more information on the genes contained in the locus. Donald et al. [82] used diverse oligonucleotide primers that targeted a conserved region of the NBS domain of resistance gene analogs (*RGA*s) to assess whether any of these were associated with the presence of the *Run1* locus. They found that three RGA markers were tightly linked to the *Run1* locus, so it was likely that *Run1* belonged to that family of proteins [80]. Powdery mildew resistance is not the only positive characteristic inherited from *M. rotundifolia* in segregating progenies; in fact, *Rpv1* (Resistance to *Plasmopara viticola*) is tightly linked to the *Run1* locus, and both traits cosegregate [83].

Later, Barker et al. [52] constructed a genetic map using a *Run1* segregating family and narrowed down its location to the linkage group (LG) 12. Anderson et al. [84] used that information to locate the *Run1* locus to chromosome 12. All these investigations were the basis for Feechan et al. in 2013 [53] to fine-map the *Run1* locus, finding several candidate genes: six full-length *MrRGAs*, two partial TIR-only *MrRGAs*, one NBS-only *MrRGA*, and one TIR-only *MrRGA*. Full-length genes were cloned into susceptible *V. vinifera* cv. ‘Tempranillo’ and cv. ‘Portan’. The response to the infection of the transformed plants of both varieties was evaluated regarding the percentage of epidermal cells penetrated by the fungus showing PCD. For those evaluations, a genotype from pseudo-backcrossing that carried the complete *Run1* locus was used as a positive control. Based on these analyses, it was concluded that *MrRGA10*, which encodes a full-length resistance protein of the type TIR–NBS–LRR, is the gene responsible for the resistance to powdery mildew at the *Run1* locus. In other words, *MrRGA10* is the *MrRUN1* gene (JQ904636.1). This study also suggested that some of the other *MrRGA* genes located within the *Run1* locus make a minor contribution toward powdery mildew resistance, as suggested by a higher PCD induction in the control genotype carrying the full-length *Run1* locus [53].

Since it has previously been described that the powdery mildew Musc4 (M group) and NY1-137 isolates are able to overcome *Run1* resistance [85], a comparison between transgenic *MrRGA10 (MrRUN*1 gene) lines and the positive control was performed. In this experiment, the resistance of both lines against the attack of isolates belonging to the A and B groups, the most widespread genetic groups in viticultural areas, was also evaluated. As expected, the Musc*4* and NY1-137 isolates broke the resistance conferred by the *Run1* locus, while resistance was maintained against groups A and B [53,69]. It is important to highlight that despite the surmounting of resistance by the Musc4 and NY1-137 isolates, *Run1*-carrying plants could still ensure a more profitable production than susceptible plants [85]. Musc*4* and NY1-137 isolates are only present in southeastern North America [86], so it is expected that they would only be a threat for production in that area [85]. Consequently, the coevolution of the Musc4 isolate with *M. rotundifolia Run1* plants has likely mutated its effector, which allows it to evade host cell recognition [53]. Moreover, other American species, such as *V. rupestris,* have shown resistance against these same strains [87] and thus can be used as complementary sources of resistance along with loci from Asia, for cultivar development.

It is known that several effector proteins are targeted to the cell nucleus, thus altering the essential components or function of the host immune response, and that the subcellular localization of their related R proteins may be affected by nuclear translocation of effectors [88,89]. NBS–LRR R proteins act as molecular switches, which activate the transcription of genes associated with the ETI host defense response. R proteins must maintain dynamic traffic between the cytoplasm and the nucleus to permit the transduction of defense signals between the two cellular compartments [89]. In barley, it has been described that the powdery mildew (*Blumeria gramini*s) resistance MLA protein, which is a Coiled–Coil (CC)–NBS–LRR receptor, is located in both the cytoplasm and the nucleus of the cell. This suggests that for the activation of the immune response mediated by MLA to occur, nuclear accumulation is necessary. Barley is not the only plant that shows this type of activation of immune responses; in tobacco, the TIR–NBS–LRR protein responsible for the immune response against Tobacco mosaic virus (TMV) is localized only in the nucleus in inoculated cells, unlike that observed in noninoculated cells, where the receptor is located simultaneously in the cytoplasm and in the nucleus. This suggests that cytoplasmic TIR–NBS–LRR R proteins may play a role in the early recognition of pathogen effectors, which would trigger the nuclear translocalization of R proteins to the cell nucleus, which is required to activate the immune response [89,90]. Based on their previous research, Feechan et al. [53] studied the cellular location of TIR–NBS–LRR *Mr*RUN1 proteins and reported that they were mostly found in the cell nucleus and to a lesser extent in the cytoplasm. From this result, it can be proposed that *Mr*RUN1 protein could have a similar function in activating the immune response as MLA protein in tobacco. Another factor that strengthens this hypothesis is that the *MrRUN1* gene produces four different transcripts (A, B, C, and D) due to alternative splicing [53]. This characteristic has also been seen in other genes that code for R proteins [91]. In the case of the *MrRUN1* gene, transcript A encodes a full-length protein; transcript B encodes a protein with a truncation of the NB–ARC domains; transcript C encodes a truncated TIR–NBS protein, while transcript D generates a truncated TIR protein. In the case of transcripts C and D, truncated proteins are generated from a premature stop codon caused by a frameshift [53]. The possibility of four proteins synthesized from *MrRUN1* raises the hypothesis that they could play different roles in the immune response or PCD signaling, depending on their structure [53]. It has been described that these truncated isoforms of R proteins may contribute by inhibiting the negative regulation of the immune response or by participating in ETI signaling [91]. Depending on the presence or absence of certain domains, the different proteins generated by alternative splicing could have different cellular locations. For example, full-length TIR–NBS–LRR *Mr*RUN1 protein (transcript A), which has a nuclear localization signal (NLS), is probably important in signaling; in contrast, the truncated TIR–NBS protein (transcript C), which does not have an NLS, could play a signaling role in the cytoplasm [53].

Later, Massonet et al. [71] described two different DNA sequences located in *M. rotundifolia* ‘Trayshed’ in the same chromosome where *Run1* was previously mapped and named them *Run1.2a* and *Run1.2b*. These two haplotypes differed in length and the number of genes located in the locus. The *Run1.2a* locus comprises 253 genes with greater abundance of inserted and duplicated events than *Run1.2b*, which included 189 genes. Among them, 37 and 24 of these genes potentially code for CC–NBS–LRR, TIR–NBS–LRR, and NBS–LRR proteins in *Run1.2a* and *Run1.2b*, respectively.

### 4.2. Run2.1 and Run2.2

*Run2.1* and *Run2.2* were described by Riaz et al. [55] in *M. rotundifolia* ‘Magnolia’ and ‘Trayshed’. These loci are located in the same genetic region of chromosome 18 and trigger the same immune responses: PCD and limitation of the growth of hyphae. However, they differ in allele sizes and race specificity. Feechan et al. [69] challenged *Run2.1* and *Run2.2* resistance with three *E. necato*r isolates, LNYM, NY19, and Musc4 from the USA. The first two strains were obtained from *V. vinifera* in New York, and the third strain was found growing on *M. rotundifolia* in Georgia. It is important to note that *Run1*, *Run2.1*, and *Run2.2* were discovered in *M. rotundifolia* [55,80]; therefore, it was suspected that their race spectrum range could be similar. *Run2.1* and *Run2.2* triggered a successful immune response when challenged with LNYM and NY19. Curiously, both alleles showed a nonidentical response against Musc4. Specifically, in the case of *Run2.1*, a weak immune response was observed for this isolate, unlike *Run2.2*, in which the resistance was overcome [69]. Based on this finding, it was suggested that *Run2.1* provides a broader spectrum of resistance than *Run2.2* and *Run1*.

### 4.3. Ren1

Unlike other resistance loci, *Ren1* was found in *V. vinifera* and not in a wild *Vitis* species. It was discovered in the Asian table grape varieties ‘Dhandzhal Kara’ [92] and ‘Kishmish vatkana’ [93], which show complete resistance to powdery mildew [94]. Hoffman et al. [59] and Coleman et al. [95] described that the resistance conferred by *Ren1* behaves as a Mendelian monogenic trait. It is located in LG 13, and the locus contains 27 genes, which encode proteins with 15 different functions. Within these genes, there are two that encode full-length CC–NBS–LRR disease-resistant proteins and nine that could code for partial-length CC–NBS–LRR proteins. However, there is no significant match of their hypothetical products with previously described sequences [95]. In addition to the genes mentioned above, other genes of the region that could code for disease-resistant functions are cinnamyl alcohol dehydrogenase (*CAD*) genes, related to lignin biosynthesis, and a Rab1/RabD small GTPase [95], which participates in membrane trafficking in the secretory pathway [96].

Of the responses triggered by *Ren1*, a reduction of 84% of the cells penetrated by the fungus has been noted. The infection stops after haustorium formation, which halts the development of primary hypha [35,59]. Other responses involved are the triggering of PCD and the formation of callose deposits at 48 hpi and the generation of ROS at 96 hpi [35].

Recently, Possamai et al. [68] described the *Ren1.2* locus, a variant of *Ren1*. It was mapped in the Caucasus *V. vinifera* cv. ‘Shavtsitka’ and is in the same chromosomal region as *Ren1*, so it was classified as a variant of the previously mentioned locus. Interestingly, the *Ren1* and *Ren1.2* loci were found in cultivars of different origins, suggesting that Caucasus grapevines have independently developed their resistance loci in the exact same location as *Ren1*. The *Ren1.2* locus provides partial resistance to *E. necator*, reducing hyphal proliferation and sporulation [68].

### 4.4. Ren2

This locus has been identified on chromosome 14 of *V. cinerea*, providing partial resistance [63,96]. *Ren*2 was tested with three *E. necato*r isolates from New York (LNYM, NY-19, and NY-131), triggering an effective immune response that detained fungal sporulation. In contrast, fungal sporulation was reduced marginally when the same evaluation was performed with the Musc4 strain. The colony formation rate was lower in *Ren2* plants than in a susceptible cultivar (Chardonnay) [69].

### 4.5. Ren3 and Ren9

*Ren3* was found on chromosome 15 [70] of the grapevine cultivar ‘Regent’. No variations were observed initially in conidia germination, haustoria, and mycelium development between *Ren3* plants and a susceptible cultivar. The first differences appeared at five dpi, when no sporulation of the fungus was observed in *Ren3* plants, whereas in a susceptible cultivar, the fungus began to reproduce. It was observed that the immune response of *Ren3* plants was based on ROS production, the generation of callose deposits, and PCD of infected epidermal cells [65].

*Ren9* was discovered by Zendler et al. [65] through a fine-mapping study of *Ren3*. Both loci are on the same chromosome but in different regions, enabling their segregation into diverse genotypes. In progeny containing either both loci separately or together, the development of the disease was slower than in a susceptible cultivar. *Ren3*, *Ren9*, and *Ren3Ren9* genotypes showed similar hyphal growth that was lower in comparison to a susceptible cultivar. The PCD among all resistant loci combinations had the same intensity, but a variation in response time was observed between loci, being apparent from four to six dpi in *Ren3* and *Ren9*, respectively. Therefore, these loci trigger resistance responses that are equally strong, and their combination has no additive effects [64].

Within the *Ren3* locus, there are four NOD-like receptor genes (NLRs); these genes participate in the recognition of effectors from pathogens. For this reason, it is believed that one of them is responsible for resistance [65]. In the case of *Ren9*, the main candidate genes to confer resistance are genes that code for LRR-like kinase, which is essential for PAMP detection [97]. Due to the type of genes that each locus contains, it is likely that they trigger different resistance mechanisms, and it may be that *Ren3* induces ETI and that *Ren9* triggers PTI [64]. The presence of PCD in *Ren9* genotypes could be explained by the recent discovery that PTI activation induces ETI responses [98,99,100], which is the type of immunity that usually triggers PCD [101].

Like other loci from North American grapevine species, the resistance provided by *Ren3* and *Ren9* was overcome by the American *E. necator* strain NY19 [67], probably by coevolution of the fungus and host. However, these loci could be used in Europe in a pyramidal fashion with loci conferring strong resistance [64].

### 4.6. Ren4

This dominant locus is native to the Asian grapevine *V. romanetii* and is located on chromosome 18 [55]. It has been proposed that it generates a broad-spectrum response against different races of powdery mildew at the postpenetration stage, producing PCD in the penetrated epidermal cells or covering fungal haustoria with callose, thus inhibiting the uptake of nutrients by the fungus [102]. The *Ren4* response time is the fastest of the resistance to powdery mildew genes and loci. For that reason, it was proposed that it is probably related to PTI or another type of barrier [54]. PTI needs a specific receptor, but to date, a specific receptor encoded by *Ren4* has not been found to confirm such a hypothesis [52,54]. The theory that the resistance provided by *Ren*4 is of the nonhost type is also not supported by the observation that not all accessions of *V. romanetii* are resistant to powdery mildew [54,103].

Among the resistance responses attributed to these genes are a minimum cell penetration rate and the almost complete absence of the development of secondary hyphae; consequently, colonies are not observed [54].

### 4.7. Ren5

This locus has been mapped to chromosome 14 in *M. rotundifolia*. It contains close to 150 genes, and it is still not clear which is/are responsible for the resistance to powdery mildew [60]. However, some genes of the locus have been described as possible candidates, including seven genes that encode NBS–LRR resistance proteins. However, it cannot be ruled out that another pathway triggers the *Ren5* resistance mechanism; for example, a gene that codes for ENHANCED DISEASE SUSCEPTIBILITY 1 (*EDS*1) is also located in the same region, which could be another possible candidate [60,104].

It has been suggested that *Ren5* acts after forming the first fungal appressorium by delaying mycelium development. In other words, it does not affect the first steps of *E. necator* infection. From 1 dpi, differences between *Ren5* and susceptible genotypes are observed. While in *Ren5-*resistant plants the development of the fungus was blocked at the first appressorium stage, the opposite happened in susceptible plants, where the fungus formed primary and secondary hyphae, following the normal course of a powdery mildew infection. Consequently, *Ren5* affected mycelium branching and the sporulation rate [60].

It is still necessary to continue the study of the immune response triggered by *Ren5* to characterize whether an increase in ROS mediates this response and PCD, as described for other *Run* and *Ren* loci.

### 4.8. Ren6 and Ren7

The loci *Ren6* and *Ren7* were identified in the Chinese species *V. piasetzkii* on chromosome 9 and chromosome 19, respectively [57]. They segregate independently of each other, and it has been seen that the *Ren6*, *Ren7*, and *Ren6Ren7* phenotypes differ from each other [57]. Both loci generated a postinvasion PCD that in the case of *Ren6* was observed beneath the appressoria of germinated spores, while in *Ren7*, PCD occurred at the stage of secondary hyphae development. Nevertheless, the two loci display different strengths or speeds of the PCD response, which is subsequently apparent in their different effectiveness in limiting hyphal development. *Ren6* produced a stronger and faster PCD response than *Ren7*, which successfully prevented the formation of secondary hyphae at 2 dpi, whereas *Ren7* did not stop fungal growth before that stage. Between 92 and 95% of the infected cells in *Ren6* plants displayed an effective PCD, contrasting with the much smaller percentage (10–20% approximately) observed in *Ren7* plants. In the case of *Ren6Ren7* plants, no differences were observed in the development of the disease compared to plants containing *Ren6* only. In addition, Pap et al. (2016) [57] compared the PCD rates caused by the *Ren6* and *Run1* loci in crosses with the same genetic background. They observed that the *Ren6* locus exhibited an even higher resistance response than *Run1*, which showed a PCD response in approximately 85% of the infected epidermal cells. Based on these data, they suggested that *Ren6* generated a faster and stronger resistance than both *Run1* and *Ren7*. A possible explanation for this differential response may be that these R loci do not recognize the same effector; R*en6* could identify a molecule that is secreted by the fungus in earlier phases of the infection than the effectors detected by *Run1* and *Ren7*. Another hypothesis is that there could be variations in these R proteins in grape cells.

### 4.9. Ren8

Zyprian et al. [66] discovered the *Ren8* locus on chromosome 18 and observed that it may overlap with *Run2* or *Ren4*. The species of origin of this locus is unknown, since it was found in genotypes generated by a large number of back-crosses, although it is suspected that it comes from a North American grape species [102]. It was suggested that this locus mainly mediated resistance against European *E. necator* isolates [66], and the testing of *Ren8* against North American strains still needs to be undertaken to evaluate if it confers race-specific or broad-spectrum resistance. More studies are needed to characterize the responses generated by *Ren8* at both the histological and molecular levels. This locus was mapped close to three genes encoding proteins related to immune responses: an NBS protein, a TIR domain-containing protein, and Smg-4/UFT3, which regulates defense responses in *A. thaliana* [105].

### 4.10. Ren10

This locus has been mapped to chromosome 2, close to the MYB-like genes that control berry color [106]. It is hypothesized that it was introgressed from a North American grape species, which could have been generated through hexaploidization or successive genome duplications [67]. Although this locus is known to decrease hyphal proliferation and sporulation, its mechanism of action is still unclear. The immune response generated by *Ren*10 has been compared with *Ren3*, and it was seen that *Ren10* triggers a more effective response than *Ren3* in reducing hyphal development and the development of reproductive structures of powdery mildew [67].

### 4.11. Ren11

Recently, a new powdery mildew resistance locus has been reported, named *Ren11* [61], which had previously been described as a quality trait locus by Ramming et al. [62] but not mapped in that research. *Ren11* is located on chromosome 15 [59]. It was first identified in the *Vitis* hybrid ‘Tamiani’, generated by the cross of *V. aestivales* × *V. vinifera* ‘Malaga’ and its resistant F_1_ offspring, called B37-28. It confers moderate to strong resistance [62], although it is unknown whether this is race-specific or broad-spectrum [61]. A significant diminution of fungal penetration, hyphal length, and microcolony formation have been observed in grapevine plants that harbor *Ren11* in comparison with a susceptible *V. vinifera* cultivar [62]. Based on the location of *Ren11*, it has been proposed that it could confer resistance to downy mildew [61] since it overlaps with the resistance loci *Rpv23* and *Rpv26* [107,108].

## 5. Locus Stacking: The Search for Durable and Broad-Spectrum Resistance

Currently, one of the main objectives of grapevine breeding programs worldwide is the development of durable and strong resistance to powdery mildew, through independent modes of action. The most important desirable outcome of such programs is that the resistance must be durable. Because grapevine plants are productive for at least twenty years, resistance needs to be maintained through that period of time [55,57]. To achieve this goal, a pyramiding strategy has been proposed, which combines various resistance loci in the same genotype [109]. To ensure the durability of this resistance, it is necessary to mix loci that have different mechanisms of action, spectrums of target isolates, and contributions (minor and major) to the resistance [83,110]. Referring to this last aspect, it is important to consider that even though initially more promising results are observed when a gene or locus with a major effect is used, this can favor the selection of isolates of the fungus that are capable of overcoming this major resistance loci [110,111], and if resistance is based only on the presence of one gene, the fungus could mutate its effector and evade immune recognition [57]. A clear example of this is what happened between the *Run*1 locus and the Musc4 isolate, which is probably due to a long coexistence with *M. rotundifolia*, the donor specie of *Run1*, which likely mutated its effector to overcome the resistance conferred by this gene [51,85]. This response has not only been observed with *Run1*; *Ren3* and *Ren9* loci resistance were also overcome by a North American *E. necator* strain, despite these loci only conferring partial resistance [67]. These results suggest that in the case of the development of new grapevine cultivars with resistance to powdery mildew, it is important to consider the origin of the genes or loci when pyramiding, prioritizing the combination of resistance sources from species with diverse geographical origins. In the case of the development of resistant cultivars in North America, the high genetic variability of powdery mildew in that area [83] is a challenge for breeders.

More studies are needed to evaluate the best combination of genes and loci for each viticultural area. Currently, the immune responses of some genotypes that have more than one source of resistance have already been characterized (Table 2). The presence of more than one resistant gene or loci does not generate a more intense resistance response in all the cases studied, demonstrating that combinations do not always generate additive effects (Table 2). This is the case of the *Run1.2a/b* genotypes that did not show any difference in PCD induction and secondary hyphae formation, compared to genotypes carrying just one of these loci [68]. Another example is the combination of *Ren3* and *Ren9*, which did not generate an immune response that has an advantage in terms of the intensity or speed of the response compared to *Ren3* alone [64]. This response has also been observed in *Ren*6*Ren*7 genotypes, which had an equal response to the *Ren6* locus alone [57]. On the other hand, the combinations of *Run1Run1.2a/b, Run1Ren1*, and *Run1Ren2* did show an additive effect, as the combination of both genes/loci generated a stronger immune response than the one triggered by each one individually. For example, *the Run1Run1.2a/b* genotypes showed less formation of secondary hyphae than each gene/locus separately [68], while in the case of *Run1Ren1* genotypes, a more intense defense response was observed in terms of ROS production, callose accumulation, PCD, and activation of STILBENE SYNTHASE 36 (*Vv*STS36) and PENETRATION 1 (*VvPEN*1) than each of them separately [35]. The STS gene family encodes stilbene synthases, which catalyse the production of the stilbenes, compounds that have antimicrobial activities in plant defense [39]. *PEN1* has a role in the traffic of secretory vesicles that could be associated with penetration resistance against powdery mildews [111]. For *Run1Ren2* genotypes, a significant decrease in colony formation was seen compared to genotypes containing only *Run1* or *Ren2* [68].

## 6. Development of Genetic Resistance by Gene Editing

As an alternative approach to identify genes conferring resistance to *E. necator*, searching for susceptibility genes (*S* genes) can be an interesting strategy since inactivation of those S genes should lead to resistance to powdery mildew. An example of these S genes is the mildew locus O (*MLO*), which is conserved throughout the plant kingdom. Loss of function of certain members of the *MLO* gene family increases resistance to powdery mildew in *A. thaliana*, pea, tomato, wheat, and pepper. In *Vitis*, the combined silencing of *VvMLO6*, *VvMLO7,* and *VvMLO11* produced a 77% decrease in *E. necator* infection [112]. However, although gene editing by Crispr–Cas9 of *VvMLO3* did lead to an increase in resistance to powdery mildew, this was only observed in heterozygous plants, as the homozygous mutation produced plant death by necrosis, which suggests a pleiotropic function of this gene in *Vitis* [113].

## 7. Final Remarks

There is a trend to decrease pesticide use worldwide, either guided by consumer preferences for food production that is safer for the environment and for human health [114] or due to restricted legislation on the allowed number of applications of permitted chemicals. These changes represent a challenge for the wine and table grape industry since most commercial cultivars are highly susceptible to powdery mildew. For that reason, it is necessary to find control strategies with a lower environmental impact and, at the same time, ensure that production is both profitable and of high quality.

The generation of grapevine cultivars that naturally carry resistance genes is a tool that could allow a reduction in pesticide use. However, it has the limitation that the processes of grapevine breeding carried out through directed pollination require several years to obtain individuals [54] with attractive, productive, and quality characteristics.

Breeding programs focused on the development of table and wine grapevine genotypes that are resistant to powdery mildew isolates are being carried out in several countries. Examples of these programs are being pursued in the Fondazione Edmund Mach in the north of Italy [115], INRA-ResDur in France [116,117], The University of California, Davis and Cornell University-USDA in the USA [118], The Commonwealth Scientific and Industrial Research (CSIRO) in Australia [118], the Research of Viticulture and Enology in Hungary [118], the Pontificia Universidad Católica de Chile together with Consorcio de la Fruta in Chile [35] and the Instituto de Investigaciones Agropecuarias (INIA) in Chile [119], and the Institute for Grapevine Breeding Geilweilerhof in Germany [119]. Some of these institutions have already registered commercial cultivars with genetic resistance to powdery mildew [116].

In recent years, interest in the research of powdery mildew resistance genes or loci has increased, which has led to the discovery of new resistance genes or loci. However, it is still necessary to study the immune response pathways of *Run* and *Ren* loci and gene families in detail at the molecular level. A better understanding of their molecular mechanism of action could help in choosing the most suitable combination of genes and loci to stack.

Although multiple genetic and histological characterizations of *Run* and *Ren* have been undertaken, studies are still needed to characterize the agronomic and physiological consequences of resistance to powdery mildew. Currently, it has not been described whether the resistance conferred by these genes or loci incurs an energy cost for the plant, for example, by alteration of the photosynthetic rate or carbon assimilation. Recently, it has been described that the immune response produced by the genes of resistance to the biotrophic fungus *P. viticola* is associated with a decrease in the photosynthetic rate of resistant grapevines [120]. Therefore, it would be interesting to study if the resistance conferred by *Run* and *Ren* genes and loci produce alterations in the physiology of the plant, as observed in *P. viticola*.

## Figures and Tables

**Figure 1 pathogens-11-00703-f001:**
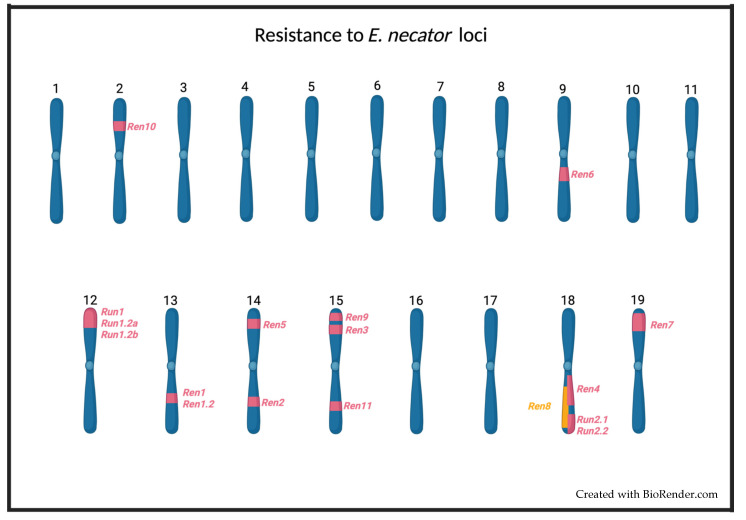
Illustration of the chromosomal location of loci of resistance to *E. necator* in *Vitis vinifera Run1–Run1.2a/b* (Chr12) [52,53,71,80,81]; *Run2.1–Run2.2* (Chr18) [55]; *Ren1–Ren1.2* (Chr13) [59,68,82,83,84]; *Ren2* (Chr14) [63,85]; *Ren3* (Chr15) [64,70]; *Ren4* (Chr18) [54]; *Ren5* (Chr14) [60]; *Ren6* (Chr9) [57]; *Ren7* (Chr17) [57]; *Ren9* (Chr15) [64]; *Ren10* (Chr2) [67]; and *Ren11* (Chr15) [61] are marked in red on the figure. *Ren8* [66] is marked in orange to highlight that it may overlap with *Ren4* and *Ren2.1–Ren2.2* [66].

**Table 1 pathogens-11-00703-t001:** Summary of powdery mildew resistance loci discovered in Vitaceae family. The origin, host response, and resistance level to powdery mildew of each locus are shown. Donor species and area of origin are also specified. In the host, the responses are programmed cell death (PCD), the production of callose, and the increase in ROSs. The level of resistance is considered as ‘total’ in the absence of visible symptoms and ‘partial’ for cases where the symptomatology decreases without disappearing completely. The variable classification was used for cases in which a race-specific response was observed, being ‘total’ for some strains and ‘partial’ for others.

Locus	Donor	Host Response			Resistance Level	Reference
		PCD	Callose	ROS		
*Run1*	*M. rotundifolia* G52 ^1^	Yes	Yes	Yes	Variable *	[35,52,53]
*Run1.2a*	*M. rotundifolia* ^1^	Yes	n.i.	n.i.	Variable *	[71]
*Run1.2b*	*M. rotundifolia* ^1^	Yes	n.i.	n.i.	Variable *	[71]
*Run2.1*	*M. rotundifolia* ‘Magnolia’ ^1^	Yes	n.i.	n.i.	Partial	[55]
*Run2.2*	*M. rotundifolia* ‘Trayshed’ ^1^	Yes	n.i.	n.i.	Partial *	[55]
*Ren1*	*V. vinifera* cv. ‘Kismish vatkana’ ^2^	Yes	Yes	Yes	Total	[59]
*Ren1.2*	*V. vinifera* cv. ‘Shavtsitka’ ^3^	Yes	n.i.	n.i.	Partial	[68]
*Ren2*	*V. cinerea* ^2^	Yes	n.i.	n.i.	Partial	[63,69]
*Ren3*	‘Regent’ ^4^	Yes	Yes	Yes	Partial	[64,70]
*Ren4*	*V. romanetii* ^2^	Yes	n.i.	n.i.	Partial	[54]
*Ren5*	*M. rotundifolia* ‘Regale’ ^1^	n.i.	n.i.	n.i.	Total	[60]
*Ren6*	*V. piasezki* ^2^	Yes	n.i.	n.i.	Total	[57]
*Ren7*	*V. piasezki* ^2^	Yes	n.i.	n.i.	Partial	[57]
*Ren8*	Unknown ^4^	n.i.	n.i.	n.i.	Partial	[66]
*Ren9*	‘Regent’ ^4^	Yes	n.i.	n.i.	Partial	[64,65]
*Ren10*	‘Seyval blanc’ ^4^	n.i.	n.i	n.i.	Partial	[67]
*Ren11*	*Vitis aestivalis* ^2^	n.i.	n.i.	n.i	Partial	[61]

^1^ North American *Vitis*, ^2^ Asian *Vitis*, ^3^ Caucasian *V. vinifera* cultivar, ^4^ Interspecific hybrids of *V. vinifera* with North American *Vitis* species, * Genetic resistance was overcome by Musc4 *E. necator* isolates [69,85], and n.i.: No information available.

**Table 2 pathogens-11-00703-t002:** Effect on resistance reported by pyramiding different loci in the same genotype. Additive effect refers to the fact that the combination of loci generated a stronger immune response compared to the effect of each locus separately.

Effect Type	Loci	Reference
Additive	*Run1Run1.2a/b*	[69]
	*Run1Ren1*	[35]
	*Run1Ren2* *	[69]
Nonadditive	*Run1.2a/bRun2.2*	[69]
	*Ren3Ren9*	[64]
	*Ren6Ren7*	[57]

* Race-specific, as this effect was not seen with the Musc4 isolate.

## Data Availability

Not applicable.

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
