# Peer review of "Powdery Mildew Resistance Genes in Vines: An Opportunity to Achieve a More Sustainable Viticulture"

_pathogens, 2022, doi:10.3390/pathogens11060703_

Round 1

Reviewer 1 Report

This manuscript presents a review of current knowledge around genes and loci  in grapevines that confer resistance to powdery mildew.

The first thing to say is that I found the English expression to be quite poor and confusing in many parts of the document which made it difficult to read. There were so many corrections required I decided to make them in the document with track changes instead of listing them.  

Although the review is reasonably comprehensive in its coverage of the published literature to do with grapevine powdery mildew resistance, too much of the review was devoted to simply repeating the observations of the published papers rather than integrating these findings and comparing them to current knowledge about powdery mildew resistance genes from other crop species.

There were also examples of what appeared to be a lack of understanding of key concepts around the structure and function of R proteins. For example in section 4.3 on two occasions the structure of the CC-NBS-LRR type resistance protein was written as NBS-LRR-CC implying the CC domain was at the C-terminal end of the protein.

There were also a number of examples of mis-interpretation of results or conclusions of published paper.

I have also added a number of questions and comments in the revised manuscript (see attached) that the authors need to address before the manuscript could be considered acceptable for publication.

Author Response

Comments and suggestions for authors: 

Grapevine (Vitis vinifera) is one of the main fruit crops worldwide. However, the yield and quality of grapevine is affected by powdery mildew, which is caused by Erysiphe necator. This article reviewed the powdery mildew resistance genes and loci, and some examples of grapevine breeding programs, which will be beneficial to grape resistance utilization.

The biggest problem of this manuscript is that the language description is to some extent a list of literature, not integrated into an organic whole, and not concise enough.  

Response 1:  We value your commentaries, as they have considerably improved the manuscript. To improve the English grammar of our manuscript we sent it to a native English speaker, who reviewed and corrected it. We expect that our manuscript is clearer now. 

Reviewer 2 Report

Grapevine (Vitis vinifera) is one of the main fruit crops worldwide. However, the yield and quality of grapevine is affected by powdery mildew, which is caused by Erysiphe necator. This article reviewed the the powdery mildew resistance genes and loci, and some examples of grapevine breeding programs, which will beneficial to grape resistance utilizaition.

The biggest problem of this manuscript is that the language description is to some extent a list of literature, not integrated into an organic whole, and not concise enough.  

Author Response

Dear reviewer: 

We value all your observations and suggestions. We attached a file with all the corrections.

Best regards.

Round 2

Reviewer 1 Report

The revised manuscript is a significant improvement on the original version but there are still a number of issues which need to be addressed (outlined in the attached manuscript) and many grammatical errors that need to be fixed.

Author Response

Dear reviewer 1: 

Thanks for your comments and suggestions. I attached a letter with the response to your comments. 

Best regards,

Viviana Sosa-Zuniga 
